# Health Technology Assessment Development in Vietnam: A Qualitative Study of Current Progress, Barriers, Facilitators, and Future Strategies

**DOI:** 10.3390/ijerph18168846

**Published:** 2021-08-22

**Authors:** Hwa-Young Lee, Thuy Thi-Thu Nguyen, Saeun Park, Van Minh Hoang, Woong-Han Kim

**Affiliations:** 1Department of Global Health and Population, Harvard TH Chan School of Public Health, Boston, MA 02115, USA; hwlee@hsph.harvard.edu; 2Institute of Convergence Science (ICONS), Convergence Science Academy, Yonsei University, Seoul 03722, Korea; 3Department of Organization and Economics of Pharmacy, University of Medicine and Pharmacy at Ho Chi Minh City, Ho Chi Minh City 70000, Vietnam; nguyenthuthuy@ump.edu.vn; 4Division of Epidemiology and Community Health, School of Public Health, University of Minnesota, Twin Cities, MN 55455, USA; park2300@umn.edu; 5Hanoi University of Public Health, Hanoi 11909, Vietnam; hvm@huph.edu.vn; 6Program in Global Surgery and Implementation Science, JW LEE Center for Global Medicine, Seoul National University College of Medicine, 103 Daehak-ro, Jongno-gu, Seoul 03087, Korea; 7Department of Thoracic and Cardiovascular Surgery, Seoul National University College of Medicine, 103 Daehak-ro, Jongno-gu, Seoul 03080, Korea

**Keywords:** health technology assessment, economic evaluation, low- and middle-income countries

## Abstract

Introduction: To make more efficient use of limited resources, Vietnam incorporated health technology assessment (HTA) into the decision-making process for the health insurance benefit package in 2014. We evaluated progress in HTA institutionalization in Vietnam based on the theoretical framework developed by the National Institute for Health and Care Excellence and the Health Intervention and Technology Assessment Program, identified negative and conducive factors for HTA development, and finally suggested policy recommendations that fit the Vietnamese context. Methods: Semi-structured in-depth qualitative interviews were conducted between December 2017 and March and April 2018 with a purposive sample of 24 stakeholders involved in decision-making for health insurance reimbursement. We employed thematic analysis to examine themes within the data. Results: Despite a variety of activities (e.g., training and advising/mentoring) and a substantial level of output (e.g., policy statements, focal points assigned, and case studies/demonstration projects), Vietnam has not yet reached the policy decision stage based on HTA with scientific integrity and active stakeholder participation. Most respondents, except some clinicians, supported the use of HTA. The lack of capacity of human resources in the government sector and academia, the limited data infrastructure, the absence of guidelines, the government’s interest in immediate budget-saving, and public resistance were identified as barriers to the advancement of HTA. Conclusions: A structured data repository, guidelines based on the Vietnamese context for both policy decision-making at the central level and daily clinical decision-making at the micro-level, and integration of a participatory process into HTA are suggested as priorities for HTA institutionalization in Vietnam.

## 1. Introduction

Since Vietnam introduced social health insurance (SHI) at the national level in 1992 [1], there has been a rapid expansion in population coverage. Specifically, the number of SHI enrollees was 10.4 million in 2000, which was equal to 13.4% of the total population. This has increased to 30.5 million in 2006 and again to 77.3 million in 2017 (35.8% and 81.7% of the total population, respectively) [2,3]. However, the government has been plagued by budget constraints due to low premiums and a low proportion of formal sector workers among enrollees. In addition, using fee-for-service compensation for the main payment method has hindered the efficient use of a limited budget. In this context, preparing a mechanism to secure financial efficiency in the insurance budget is an urgent step for Vietnam.

The World Health Organization (WHO) acknowledged the importance of health technology assessment (HTA) for successful progression toward universal health coverage (UHC) and recommended the use of HTA to inform insurance coverage decisions and the development of clinical guidelines [4]. HTA is defined as a multidisciplinary process involving a wide range of capacities that summarizes information about the characteristics, short- and long-term effects, and/or impacts of health technologies and interventions [5]. While high-income countries (HICs) have long used HTA in decision-making, developing countries have only recently recognized the need for HTA while pursuing UHC. Vietnam introduced HTA in 2014 and has made significant efforts to facilitate HTA institutionalization since then [6]. However, there have been several challenges.

Although developed countries’ experiences with the development or use of HTA systems have been relatively well documented and disseminated through academic articles and policy reports [7,8,9,10,11,12], developing countries’ experiences have not been widely shared. Most reports are simply descriptions of barriers without an accurate evaluation of their status or progress. A handful of studies have evaluated the scope and quality of health economic evaluations performed in Vietnam [13,14,15]. However, no studies have examined the barriers or perceptions of HTA development in Vietnam.

Thus, this study aimed (1) to evaluate the progress that Vietnam has made in HTA development in light of the HTA development framework in low- and middle-income countries (LMICs) developed by the Health Intervention and Technology Assessment Program (HITAP) and the National Institute for Health and Care Excellence (NICE) International; (2) to identify hindering or facilitating factors for HTA development, and; (3) finally, to suggest real-world recommendations based on interview findings and review of other countries’ experiences to design optimal HTA for the Vietnamese context.

### A Conceptual Framework for HTA Development in LMICs

The HITAP and NICE International developed a conceptual framework to guide strategies and activities for HTA development in LMICs, as presented in Figure 1.

The technical capacity can be developed through knowledge transfer by training and advising/mentoring individual researchers. Capacity building is critical for instilling confidence and for maintaining momentum in evidence-based policy decision. Additionally, convening stakeholder-participatory processes and securing financial resources need to be initiated. However, these activities involve efforts from a higher level such as organizations or local or central governments. Then, deliverables such as HTA policy statements; resources committed by the government; guidelines; demonstration projects; and trained officers, researchers, and policy makers are expected to be produced. In the next step, HTA-informed policy decision will be expanded and regularly made based on these outputs as long as conducive factors such as political commitment among policy leaders to progress to UHC and to use evidence and tools to achieve it, stakeholder participation, and the scientific integrity of HTA are sustained [6].

## 2. Methods

### 2.1. Study Design

An evaluation of progress in HTA development was conducted using the triangulation method by blending and integrating multiple sources of data such as interviews, policy documents, and the literature. On this basis, semi-structured qualitative research was employed as the primary tool in this study to achieve our main research aim of identifying facilitators and barriers to HTA development.

### 2.2. Study Participants

Participants were purposively selected by a combination of the snowballing technique and maximum variation sampling. Key informants assisted in the identification of potential participants. Specifically, participants were referred by other participants if they were expected to have the ability to provide relevant information and to have accessibility [16]. We used maximum variation sampling to seek representativeness that may be lacking in convenience sample by including a wide range of extremes. It is based on the principle that, if one deliberately tries to interview a very diverse selection of people, their aggregate answers can approximate representativeness of the entire population [16]. Therefore, we tried to include stakeholders working in clinical practice (who are most likely to be negative to HTA introduction), government officers (who are likely to be positive), as well as researchers (who are most likely to be neutral). Additionally, special efforts were made to include decision-makers working at the macro-and meso-levels and in both the northern and southern regions of Vietnam.

### 2.3. Data Collection

In-depth interviews were carried out by three researchers during two time periods (December 2017 and March and April 2018). All interviews were performed at locations convenient to the interviewees in English or Vietnamese. Interviews performed in Vietnamese were translated to English later by two Vietnamese researchers who are fluent in English. To ensure reliability of the translation, the two Vietnamese researchers cross-checked the translation results.

We began each interview with several short questions about the participant’s demographic information, employment characteristics, and current involvement in HTA related to health insurance decision-making. Then, semi-structured questions about current HTA use, barriers to HTA development in Vietnam, and possible strategies to overcome barriers were asked (Interview questionnaire is available in Appendix A). Questions about the status in HTA development were structured based on the framework previously provided as well as by a search of the gray literature, research papers, or policy reports. Respondents were allowed to discuss other issues that were not on the questionnaire freely if those issues were relevant and important to the topic.

### 2.4. Data Analysis

The interviews were transcribed verbatim and coded following the guidelines of Strauss and Corbin (1990) [17]. The constant comparative method was employed to explore features of experience or opinions; in this process, analytic categories were inductively established by repeatedly comparing and checking items with the rest of the data [18]. Examining the current decision-making mechanism and how far it has progressed in HTA development requires fact-checking to validate the evidence elicited from the interviews. We confirmed the responses from one interviewee with another interviewee. When responses from two interviewees were not consistent, source triangulation (i.e., comparing data from different qualitative methods such as reviewing documented evidence) was employed [16]. Inconsistent responses for which we could not find written evidence were discarded.

The evaluation of the progress in HTA development was based on the conceptual framework developed by the HITAP and NICE international. The policy suggestions were elicited by a combination of interviews, reviews of other countries’ experiences, and the authors’ opinions based on study findings. Thematic analysis was conducted through a series of content clustering. Specifically, the themes and subthemes were developed as follows: First, the analysis started with open coding. Two researchers read all transcripts line by line in parallel, and then, relevant sentences were coded. Subsequently, similar codes were grouped into analytical categories whereby descriptive themes and subthemes were generated. The coding results were cross-checked by the two researchers, and different properties of these categories and the relationships between them were discussed. Pseudonyms were utilized to ensure the anonymity of interviewees. Representative quotes for each theme or subtheme were presented to illustrate themes, with minor revisions to protect respondents’ anonymity or to correct the grammar.

## 3. Results

### 3.1. Participants

Twenty-four participants were recruited (Table 1).

### 3.2. Current Decision-Making Regarding the Health Insurance Benefit Package

Decision-making regarding the health insurance benefit package in Vietnam has been heavily biased toward clinical effectiveness, relying on clinicians’ opinions. Consequently, almost all health technologies with even a slight clinical efficacy have been listed in the benefit package without consideration of the comparative effectiveness and efficiency. For example, as of December of 2017, the drug formulary included 845 active substances and approximately 1400 drugs in total. Drug expenditures also accounted for approximately 39% of the total health insurance fund.

In early 2017, Vietnamese MoH increased co-payments for the drugs in the benefit package, but clear evidence for the rate of increase was not provided, as phrased by one interviewee.


*“The osteoporosis drugs, one of the 20 highest consumed in Vietnam, had some changes, from 100 percent coverage to a co-payment of 50:50 percent. However, this was done without any information, (without) any research.”*
(female respondent in her 30s from academia)

At the hospital level, drugs are included in the prescription list through the bidding process as stated in Circular No11/2016/TT-BYT on regulating the bidding process for drugs at medical facilities (dated 05/11/2016 and issued by the Ministry of Health). They use a formula named “ABC/VEN” recommended by VSS to prioritize the procurement and storage of drugs (Table 2). The ANC/VEN criteria are stated in Circular No. 21/2013/TT-BYT regulating the organization and operation of the Drug and Treatment Council in hospital (dated 08/08/2013 and issued by the Ministry of Health). Hospitals consider the severity of the target disease, drug prices, and a few other factors including the health insurance reimbursement status and manufacturer quality during the bidding process.


*“We prioritize brand-name drugs for severe or chronic diseases such as diabetes, cardiovascular disease, cancer, and for diseases (treated) in the ICU. For other mild diseases, we use generic drugs, especially Vietnamese drugs with low prices. In general, we have a recommended usage percentage between generics and brand-name drugs. The Department of Health in Ho Chi Minh City recommended that the percentage of expenditures for brand-name drugs account for 25% of the whole budget, and now we just maintain that ratio.”*
(female respondent in her 30s from a hospital)

### 3.3. Progress in HTA Development in Vietnam

#### 3.3.1. Activities

International organizations have continuously provided education and training for central and local government staff as well as researchers working for universities or non-governmental organizations (e.g., the Health Economic Association). The HITAP and NICE were the most active supporters of this initiative. International organizations also provide promising researchers with grants to pursue academic degrees related to HTA.


*“Now we have three researchers who are receiving training abroad with grants. One is at a university in the Netherlands for a master’s in health economics. The other two are in Thailand for master’s training in health technology assessment. Recently, we are recruiting for master’s degree students at Mahidol University for next year. Those international training programs are funded by IDSI (the International Decision Support Initiative) project and also by HITAP as well.”*
(female respondent in her 30s from the government)

HITAP and NICE International provided advise and mentorship, working very closely with the Health Strategy and Planning Institute (HSPI), a Ministry of Health (MoH)-affiliated research institution in Vietnam assigned as a focal point of HTA development.

As a multi-stakeholder process, the pharmacoeconomic committee was newly established as an advisory body under the Department of Health Insurance within the Vietnamese MOH in April 2017 composed of 11 members from academia and government agencies. Their standing responsibility is reviewing the HTA documents submitted and working on making guidelines. However, only half of the members actively participate in this process, and the identity of the committee has yet to be clearly established. Therefore, the leverage of the committee is not yet stable or well-recognized, as described by one interviewee.


*“We had a sort of meeting to set about how to function, how often we meet. However, our job description has not been fixed yet.. it’s not that clear about the function or our value.”*
(male respondent in his 40s from academia)

As for resource mobilization, HSPI has been able to obtain international grants for HTA quite steadily, mainly from Atlantic Philanthropy and the Rockefeller Foundation. The MoH also provides regular internal funding, but that funding is insufficient to perform HTA research or develop human resources. Therefore, the HSPI has used its budget in a strategic way, whereby regular internal funding from the MoH is allocated to develop new research topics for obtaining external grants while external grants are used for practical research.

#### 3.3.2. Output

The integration of HTA into priority setting for the health insurance benefit package was a formal decision at the policy level stated in Circular 30 on the list and rates, payment conditions for pharmaceutical chemicals, biological products, radioactive drugs, and markers within the scope of benefits of health protection participants (Circular No.30/2018/TT-BYT dated 10/30/2018 and issued by MoH). The HSPI, which was established in 2013 as an institute dedicated to evidence-making, providing consultation, training, and collaborating with international partners in the field of health policy and the healthcare system, acts as the focal point of HTA development. It has six departments, of which the Department of Health Economics is responsible for HTA work. The department of Health Economics of HSPI has been involved in various tasks to develop HTA such as providing consultation and feedback to the review of HTA-related dossiers by MoH, developing economic evaluation research projects and performing them with funds from international organizations, organizing undergraduate and postgraduate training on HTA, establishing and maintaining relationships with international organizations, etc. They performed pilot cost-effectiveness studies for three items between 2014 and 2016 with technical support from HITAP: (1) peginterferon alfa-2b or alfa-2a with ribavirin for hepatitis C [19], (2) magnetic resonance imaging (MRI) services (not formally published), and (3) trastuzumab for metastatic breast cancer (not formally published). The results of the study on MRI services led to policy changes regarding MRI test indications.

However, progress in several important output domains has not been made. There are two important guidelines in HTA development: technical guidelines on how to conduct and report HTA, and guidelines on how to integrate HTA results into decision-making. The original plan of the Vietnamese MoH was to finish developing the technical guideline by 2017, focusing on medicines and consumable medical supplies, and to begin to work on the latter from 2018. However, the entire timeline was delayed. Furthermore, despite continuous external and internal support for training HTA experts, academic researchers with competencies in economic evaluation modeling and government staff able to interpret and utilize HTA results for policy were still far from sufficient.

#### 3.3.3. Intermediary Outcomes

At the time of the interviews, submission of HTA from drug manufacturers was only a recommendation, not a binding requirement. Consequently, the quality of evidence submitted from manufacturers varied. Few submitted cost-effectiveness evidence. Some submitted evidence from systematic reviews of clinical efficacy or cost-effectiveness performed in other countries. Most companies submitted only budget impact analysis results. The government was concerned more about the immediate impact on the budget than long-term cost-effectiveness, which has undermined the motivation to produce high-quality evidence. Stakeholder participation was also very limited. The public involvement mechanism was not in place yet.

### 3.4. Perception of HTA

A preponderance of interviewees recognized the value of HTA. Only a minority of interviewees, mostly clinical experts, showed negative attitudes toward the use of HTA in decision-making. They expressed mistrust in technical concepts, mainly due to a poor understanding of HTA as explained below.


*“Cost-effectiveness analyses rely on models. Like Markov models. Policymakers, especially clinicians, don’t like calculation very much. They are not very clear about it. They say ‘Oh, you will do some trick, you will do some calculation, not a real one’.”*
(male respondent in his 50s from academia)

They also denied the feasibility of utilizing HTA in decision-making because it is too complex to understand.


*“There was some disagreement between pharmacoeconomic people and clinical people…. Pharmacoeconomic people, we have the same understanding of HTA and data. However, clinical people, they think this complex concept is unnecessary. They think it is not feasible.”*
(female respondent in her 30s from the government)

All of the interviewees who supported the use of HTA agreed that it should be expanded to other areas beyond drug listing in the formulary. Preventive programs, medical equipment, and medical supplies were mentioned as examples. However, opinions on when to mandate HTA submission diverged. Some insisted on the rapid uptake of mandatory HTA.


*“HTA for decision-making on reimbursements should be mandatory soon because HTA is clear and transparent and everyone can follow it very easily. And (it involves) no corruption and no lobbying. Not much lobbying”*
(male respondent in his 50s from academia)

Others preferred to wait until infrastructure regarding HTA is fully equipped.


*“I think in our condition of lacking data, we should just encourage (not mandate) HTA for the time being before making it formal.”*
(female respondent in her 40s from a hospital)

### 3.5. Barriers to the Use of HTA

Various obstacles to HTA development in Vietnam emerged from the interviews.

*Lack of capacity:* Lack of capacity, especially for those involved in decision-making, was emphasized as the biggest barrier to the development of HTA in Vietnam. Researchers also lacked expertise in modeling and communication skills about the HTA results with stakeholders. While pharmaceutical companies usually outsource HTA preparation for their product to external researchers in universities or consulting firms, the capacity of researchers does not meet the demand.


*“Not many researchers here in Vietnam can do health economic evaluation. I know everyone who can do… maybe fewer than ten people. Especially, modeling… very few, fewer than five people can do modeling.”*
(male respondent in his 40s from academia)


*“Some medical universities or pharmacology universities have centers for HTA research. But, even if they are universities, their capacity is still limited.”*
(male respondent in his 40s from other organization)

*Lack of data:* Since cost-effectiveness is highly sensitive to even small changes in the effectiveness or cost parameters, accurate and unbiased data are critical to the integrity and reliability of HTA. For this reason, HTA requires extensive literature reviews using systematic methods to provide an exhaustive summary of current evidence relevant to health technology. However, most LMICs, including Vietnam, rarely have adequate local data. They also do not have enough resources for purchasing subscriptions to existing databases.


*“We have limitations in data resources… epidemiology data, disease data, prevalence, incidence…everything we don’t have, also costing data and willingness to pay…no.”*
(female respondent in her 30s from academia)

*Absence of guidelines:* Interviewees stated that clear guidelines would improve the uptake of HTA.


*“There is no standard procedure pharmaceutical companies can follow. So they submit different documents, different levels of information, and different levels of evidence.”*
(male respondent in his 40s from academia)

*Insufficient government commitment:* Economic evaluation aims at long-term efficiency rather than immediate cost-saving. However, the government is more interested in immediate fiscal savings, partly due to the pursuit of performance and partly due to current financial constraints.


*“They (government staff) say ‘I am the director now and I care about the budget right now and I do not care about the budget 15 years later.”*
(male respondent in his 50s from academia)


*“For example, for the treatment of hepatitis C, we reviewed if the health insurance (was willing to) spend more money for patient treatment to save costs for the treatment of liver cancer and cirrhosis occurring in ten years later (from preventing hepatitis C). They know (the benefit) but they have budget constraints right now so they can’t (allow) more expenditures today to save the money for a future treatment.”*
(male respondent in his 50s from academia)

*Resistance from consumer:* The possibility of deprivation of current benefits provokes patients’ resistance.


*“For example, uh.. the health insurance… they would like to cut the budget for the biologic treatments because they are not very cost-effective. I think that’s reasonable, but they cannot. The reason they explain to us is because, in the newspaper, some interest group write a lot of letters and they say “I’m now on that treatment and if you cut the treatment, it means you will kill me” or something like that… You can imagine.”*
(male respondent in his 50s from academia)

### 3.6. Facilitators of the Use of HTA

*Training and education of government staff:* The interviewees agreed that education and training for capacity building, especially for government staff directly or indirectly involved in decision-making, is the most urgent.


*“If government had enough money to do all the HTA on its own, it would be the most transparent option. When I look at other countries’ models, very few countries can pay 100% of HTA studies. So the important thing is whether we (government) have personnel with good qualifications to assess the results of HTA (submitted from manufacturers of health technologies).”*
(female respondent in her 30s from academia)

Interviewees from government organizations preferred short-term courses because long-term training might interfere with their main job. They also suggested awarding some form of official recognition, such as certificates, which can be of help for their future career.

*Connection between stakeholders:* The connection between stakeholders was highlighted as important.


*“We should make connections between researchers, policymakers and even suppliers. These connection should be more regular and larger-scale in the future.”*
(male respondent in his 50s from the government)

*Establishing reliable database:* Interviewees emphasized the importance of investment in establishing a reliable database. There has been some progress such as the computerization of health insurance data.


*“Currently, costs are sent daily to Vietnamese Social Security from about 97–98% of nationwide facilities, approximately 12,000 clinics throughout the country. This is a good data resource for future HTA in Vietnam. At least it is better than before, when HTA researchers only had small and empirical data sets collected directly in hospitals.”*
(female respondent in her 20s from academia)

However, important information is still missing, as one interviewee phrased it.


*“Vietnamese Social Security only has medial expenditure data covered by healthcare funds. We don’t have data on expenditures paid by patients and other parties. We are requesting to add non-insurance expenditures to the central database, but most hospitals still do not agree.”*
(male respondent in his 40s from the government)

*Developing guidelines and cost-effectiveness thresholds:* Guidelines carrying regulatory binding force were emphasized to facilitate the integrity and standardization of HTA.


*“We should set a standard for quality of research, issued by the MoH or relevant organizations so that we can check the quality of research to be more confident in applications.”*
(female respondent in her 30s from hospital)

About a third of participants agreed that the WHO recommendation for the threshold of a cost-effectiveness ratio, according to which a cost per disability-adjusted life-year (DALY) averted by less than three times the GDP per capita is considered to be cost-effective [20], can serve as a reasonable criterion for Vietnam. The remaining two-thirds of participants argued that Vietnam needs to find its own threshold. Regarding the flexibility of the threshold, only three participants supported adopting a clear-cut threshold.


*“It should be a fixed willingness to pay, a very clear and simple threshold to prevent controversy..… I know, not many people have that kind of thinking. I think that among the health economics (people), about only 10–15% like this idea.”*
(male respondent in his 50s from academia)

The majority of participants were in favor of using a flexible threshold depending on the target population or disease, arguing that a rigid threshold may result in discrimination against specific subgroups.

## 4. Discussion

Our assessment of Vietnam’s progress in HTA development revealed that, despite extensive activities conducted inside and outside Vietnam and the production of substantial output, some types of outputs such as guidelines and trained human resources were still in their beginning stage. Taken together, we can conclude that Vietnam was in the output-producing stage and had not reached the stage of intermediary outcomes at the time of the interviews. Some progress has been made since then. In 2018, the submission of budget impact analysis results became mandatory for new drugs intended for inclusion in insurance benefit coverage although the submission of cost-effectiveness analysis documents remained optional. In the same year, the Department of Health Insurance of the MoH proposed drafts of two guidelines: how to conduct and report the pharmacoeconomic research and how to appraise the research dossier submitted. Although they have not yet been formally published, Vietnamese experts have been using them informally. Furthermore, a network of pharmaco-economists was established in October 2019 to seek cooperation among experts although the network of pharmaco-economists has not been engaged in active tasks. Overall, they are still in the transition from the output-producing stage to the intermediary outcome stage despite the progress. The limited use of HTA despite positive perceptions might be due to barriers such as a lack of human resource capacity, limited data infrastructure, absence of guidelines, insufficient government commitment, and public resistance.

The qualitative findings from the present study validate, complement, and extend the results of previous studies. A few shared hindering factors to the use of HTA were observed in HICs. Specifically, lack of time, insufficient financial resources, and public resistance were identified as barriers in the Netherlands. Government workers’ mistrust in HTA due to a lack of understanding and a greater influence of clinicians in decision-making was also observed in the UK and Sweden [21,22,23,24]. Clinicians tend to want to adopt new and expensive high-technology, attaching greater value to clinical effectiveness than efficiency and making decisions based on individual patients rather than from the population perspective [25]. However, some distinctive phenomena compared with HICs were identified in Vietnam. Regarding methodological issues, Vietnam’s challenge mainly relates to generating HTA evidence, while concerns in HICs are more about how to apply HTA results in decision-making. Furthermore, in HICs, concerns were raised about the possibility that results might be biased in favor of the manufacturers because HTA is prepared or funded by manufacturers, while such concerns were not articulated in Vietnam. This is probably because of the new regulation introduced in 2017 that if a pharmaceutical company wants to submit a HTA dossier, it has to be submitted not by pharmaceutical companies but only by hospitals or patient associations to avoid bias. However, it is not clear whether changing the submitters can completely block leverage from pharmaceutical companies.

A lack of human resources and data sources were more prominent barriers in LMICs than in HICs. For example, Latin American countries [26], Argentina [27], India [28], and Ghana [29] reported a lack of capable experts and reliable local data as the biggest challenges, as was the case in Vietnam. Although those papers did not provide detailed information about countries’ progress with HTA development, they seemed to be in more of a beginning stage than Vietnam. For example, it was reported in Ghana that many interviewees had a misunderstanding of health technologies, associating the concept with the use of mobile phones, computers, or tele-medicines [29], whereas the Vietnamese stakeholders in our study had accurate knowledge of its definition.

### Policy Suggestions for Research and Practice

Recommendations for facilitating institutionalization of the HTA system were elicited from the present findings in addition to relevant literature. First, despite the importance of policymakers’ and government staff’s capacity to interpret HTA results with a full understanding and without misconceptions, they often find it difficult to allocate dedicated time to learning HTA because they are too busy with their routine work. Considering this, a structured data repository would be a helpful tool where all relevant information (e.g., theoretical knowledge on HTA or case studies of decision-making based on HTA) would be stored and easily consulted by government stakeholders at their convenience. The content would not necessarily have to be academic. Practical and easy-to-apply information is also valuable. The UK has a similar system named the UK NHS-EED database, which was made to assist decision-makers with interpreting and applying economic evaluation results in day-to-day decision-making. Although the UK system only limited data collection to the economic evaluation results [30], expanding its scope to include educational materials would be more beneficial for LMICs.

Second, clinical experts are involved in decision-making at multiple levels, such as at the macro-level as members of advisory committees for decision-making; at the meso-level such as local authorities or hospitals; or at the micro-level such as choosing drugs to prescribe, and/or treatment or diagnostic procedures to provide in their daily practice. Therefore, making them aware that saving resources from their efficiency-conscious decisions can provide huge benefits to others, such as expanding accessibility of services to more people and raising the quality of services, and can be a strong motivation to change their behavior. To achieve this goal, comprehensive education on HTA needs to be disseminated through multiple channels, among which integration of the content into the medical curriculum is highly recommended [31,32].

Third, technical HTA guidelines developed by HICs need to be adapted for them to be applicable to the Vietnamese setting. For example, the obligation of evaluation from a broader social perspective, which requires wider costs and benefit data outside the health sector, would not be feasible considering Vietnam’s data infrastructure and evidence-making capacity.

Fourth, a guideline on the utilization of HTA results needs to be prepared for both policy decision-making and for day-to-day decision-making in clinical practice at the micro-level. Evidence-based guidelines can encourage physicians to be responsible for efficient and effective care for their patients.

Finally, identification and interpretation of what constitutes benefits and harms of health technology is not only a technical matter but also a process involving value judgments in the relevant society [11]. A participatory process involving various stakeholders including lay people such as patients and citizens in HTA-related policy is a way of ensuring informed, transparent, and legitimate decision-making [9,33]. Although HTA experts are concerned about lay people’s lack of understanding of HTA, the ultimate goal of the participatory process is not to seek an immutable correct answer but to identify the value that society attaches to health technology. Furthermore, a mechanism to inform lay people of the knowledge necessary to understand HTA can be incorporated into the participatory process. The participatory process is required not only from a normative perspective, as the general population is a contributor to the financing of health insurance, but also for effective implementation because insights into the concepts and benefits of HTA would increase people’s acceptance of losses of their current benefits (e.g., coverage of expensive but less effective medicines) that might occur by adopting HTA.

There are several limitations to be noted in our study. Even though we made conscious efforts to comprehensively include stakeholders working at the macro- and meso-levels and working in clinics, the government, and academia, we recognize that our small, non-random sample is not likely to be representative. We could not use systematic sampling to secure representativeness because the pool of interviewee candidates from which we can select the sample systematically is so small given that HTA development is at an early stage in Vietnam. The risk of selection bias is another concern. It is possible that people with a greater interest in this topic were selected, which might have overestimated the positivity of perceptions toward HTA use in decision-making. Despite these limitations, the strengths of this study are that it is the first study to present details regarding the efforts to institutionalize HTA in Vietnam and provides practical strategies based on the context of LMICs.

## 5. Conclusions

Various barriers continue to inhibit HTA from being institutionalized as a solid decision-making mechanism. Comprehensive strategies suited to the Vietnamese context ranging from the macro- to micro-levels are needed to address those barriers. It is hoped that this study can serve as a helpful reference for assessing problems and identifying solutions in other developing countries with similar contexts as Vietnam.

## Figures and Tables

**Figure 1 ijerph-18-08846-f001:**
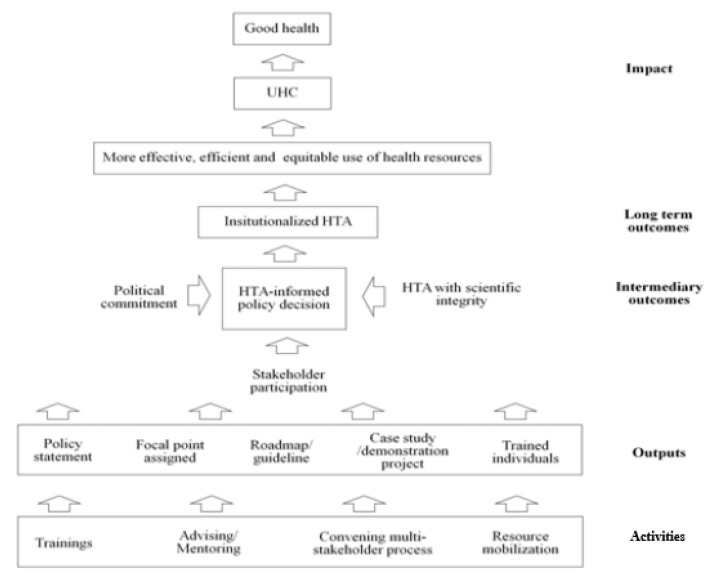
Conceptual framework of HTA development [6].

**Table 1 ijerph-18-08846-t001:** Basic characteristics of respondents.

Categories	N/%
Male/female	11/13 (45.8%/54.2%)
Age (years)	
20–29	2 (8.3%)
30–39	9 (37.5%)
40–49	5 (20.8%)
50–59	7 (29.2%)
60+	1 (4.2%)
Occupational setting	
Academia	9 (37.5%)
Government	8 (33.3%)
Hospital	6 (25.0%)
Other organization	1 (4.2%)

**Table 2 ijerph-18-08846-t002:** ABC/VEN criteria for formulary management in hospitals.

ABC	VEN
A: Drugs that account for 70–80% of the budget with 20% of Qty *	V: Vital
B: Drugs that account for 10–20% of the budget with 30–40% of Qty	E: Essential
C: Drugs that account for 5–10% of the budget with 50–60% of Qty	N: Non-essential

Note: Group AN is subject to restriction of the prescriptions, and group AE should be replaced by cheaper drugs. * Qty: Quantity.

## Data Availability

Not applicable.

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
