# Peer review of "Health Technology Assessment Development in Vietnam: A Qualitative Study of Current Progress, Barriers, Facilitators, and Future Strategies"

_ijerph, 2021, doi:10.3390/ijerph18168846_

Round 1

Reviewer 1 Report

Thank you for the opportunity to review your paper. The topic is interesting and the study is well-designed. However, I have the following comments.

Minor comments

  1. Figure 1 - the conceptual framework needs to be further explained
  2. It would be useful to make available the semi-structured interview protocol - for instance in the supplementary material
  3. Verbatim quotations extracted from interviews are very interesting; however, they should be listed and presented in a table and also grouped by emergent themes

Major revisions

  1. Methodology - as the authors employed a convenient sample, my major concern regards the process of stakeholder identification or mapping. Did the authors use established methods, such as the one promoted by WHO (i.e. systematic mapping)?
  2. I am concerned with the sentence "participants were referred by other participants if they were expected to be able to provide relevant information" - what about systematic methods of mapping especially when facing with a very small number of actors engaged (only 24).
  3.  Results and discussion/conclusions - I would prefer to see a clear link between study findings and the conceptual framework showed in the introduction. It would be necessary to better explain how study findings can be interpreted and/or view taking into consideration the established conceptual framework showed in the introduction.

Reviewer 2 Report

  1. What is the Topic list of the interviews and how was it elaborated?
  2. I did not understand the meaning given by the Vietnamese authorities to the term HTA when it was introduced.
  3. Line 58 HTA is a living concept, in full evolution. What is the reason why the authors refer to a definition from about half a century ago ??? ”HTA is defined as a policy research approach that examines the short- and long-term social consequences of the use of technology. source 5”
  4. According to the authors, the Vietnamese authorities adopted HTA in 2014 and subsequently made efforts to implement it ??
  5. line 63 The authors consider that HTA should be a formal decision-making mechanism. This is wrong!!!
  6. line 165 This process is legislated somehow or it is a discovery as a result of the interview?
  7. line 207 If the funds were insufficient for research and human resources, the question is "for what activity were they sufficient"? What was done with them for the legislative HTA process?
  8. Line 214 ”Circular 30” Souce? Year? Issued by..?
  9. Line 218 - What did the HTA department do for 5 years?
  10. Line 219 the source of the three mentioned research is missing.
  11. only budget impact results – correct to budget impact analysis results.
  12. Line 370 In 2018, HTA submission 370 became mandatory. Was it optional in 2014? Is it mandatory without the mentioned guidelines?
  13. Line 396 ”regulation introduced in 2017” ??? Unclear! Before formal HTA?
  14. Line 409 Policy suggestions for research and practice. it is a combination of general reviews with very few sources and personal opinions that are not supported by arguments even if they can be correct.
  15. Line 413 ”they often find it difficult to allocate dedicated time to learn HTA.” This statement is unclear to me. Only 8 government officials were interviewed. Are there those who are dedicated to the HTA process? In fact, only their response was relevant in terms of training.
  16. Line 458 ”it remains uncertain whether our small, non-random sample is representative” Why is uncertain? Is certain as is a non-random sample.

Round 2

Reviewer 1 Report

Well done for addressing my concerns

Reviewer 2 Report

I have no additional comments.